

**A revisit of parametrization of summer downward longwave**
**radiation over the Tibetan Plateau from high temporal resolution**
**measurements**
Mengqi Liu[a,c], Xiangdong Zheng[d*], Jinqiang Zhang[a,b,c] and Xiangao Xia[a,b,c*]
[a] LAGEO, Institute of Atmospheric Physics, Chinese Academy of Sciences, Beijing,
100029, China
[b] Collaborative Innovation Center on Forecast and Evaluation of Meteorological
Disasters, Nanjing University of Information Science & Technology, Nanjing 210044,
China
[c] College of Earth and Planetary Sciences, University of Chinese Academy of
Sciences, Beijing, 100049, China
[d] Chinese Academy of Meteorological Sciences, Beijing, 100081, China
Corresponding author:
Dr. Xiangdong Zheng
Chinese Academy of Meteorological Sciences, Beijing, 100081, China
Email: xdzheng@cma.gov.cn
Phone: 86-10-58995272
Dr. Xiangao Xia
LAGEO, Institute of Atmospheric Physics, Chinese Academy of Sciences, China
Email: xxa@mail.iap.ac.cn
Phone: 86-10-82995071





## Abstract

The Tibetan Plateau (TP) is one of hot spots in the climate research due to its
unique geographical location, high altitude, highly sensitive to climate change as well
potential effects on climate in East Asia. Downward longwave radiation (DLR), as a
key component in the surface energy budget, is of practical implications for many
research fields. Several attempts have been made to measure hourly or daily DLR and
then model it over the TP. This study uses 1-minute radiation and meteorological
measurements at three stations over the TP to parameterize DLR during summer
months. Three independent methods are used to discriminate clear-sky observations
by making maximal use of collocated measurements of downward shortwave and
longwave radiation as well as Lidar backscatter measurements with high temporal
resolution. This guarantees a reliable separation of clear-sky and cloudy samples that
favors for proper parameterizations of DLR under these two contrast conditions.
Clear-sky and cloudy DLR models with original parameters are firstly assessed. These
models are then locally calibrated based on 1-minute observations. DLR estimation is
notably improved since specific conditions over the TP are accounted for by local
calibration, which is indicated by smaller root mean square error (RMSE) and larger
coefficient of determination ($R^2$). The best local parametrization can estimate
clear-sky DLR with RMSE of 3.8 $W \cdot m^{-2}$. Overestimation of clear-sky DLR by
previous study is evident, likely due to potential residue cloud contamination on the
clear-sky samples. Cloud base height under overcast conditions is shown to be
intimately related to cloudy DLR parameterization, which is considered by this study
in the locally calibrated parameterization over the TP for the first time.




# 1 Introduction

Downward longwave radiation (DLR) at the Earth's surface is the largest component of the surface energy budget, being nearly double downward shortwave radiation (DSR) (Kiehl and Trenberth, 1997). DLR has shown a remarkable increase during the process of global warming (Stephens et al., 2012). This is closely related to the fact that both a warming and moistening of the atmosphere (especially at the lower atmosphere associated with the water vapor feedback) positively contribute to this change. Understanding of complex spatiotemporal variation of DLR and its implication is essential for improving weather prediction, climate simulation as well as water cycling modeling. Unfortunately, uncertainties in DLR are considered substantially larger than that in any of the other components of surface energy balance, which is most likely related to scarce DLR measurements with high quality (Stephens et al., 2012).

The 2-sigma uncertainty of DLR measurement by using a well-calibrated and maintained pyrgeometer is estimated to be 2.5% or 4 W m$^{-2}$ (Stoffel, 2005). However, the global-wide surface observations are very limited, especially in those remote regions. DLR is extensively estimated by proxy meteorological measurements of synoptic variables. It has been known for almost one century that the clear-sky DLR is determined by the bulk emissivity and effective temperature of the overlying atmosphere (Angstrom, 1918). Since these two quantities are not easily observed for a vertical column of the atmosphere, clear-sky DLR is alternatively parameterized as a function of air temperature and water vapor density, assuming that the clear sky radiates toward the surface like a grey body at a screen-level temperature (the standard level of meteorological measurements, generally 1.5 m above the ground). Dozens clear-sky DLR models have been developed by parameterization of different clear-sky effective emissivity ($\varepsilon_c$) to the screen-level temperature ($T_a$) and water vapor pressure ($e$). Exponential function (Idso, 1981) or power law function (Brunt, 1932) have been widely used to depict the relationship of $\varepsilon_c$ to $T_a$ and/or $e$. The coefficients of these functions are generally derived by a regression analysis of collocated measurements of $T_a$, $e$ and DLR. Most of these proposed parameterizations are thus



empirical in nature and only specific for definite atmospheric conditions. Brutsaert
(1975) was the first to develop a physically rigorous model of clear-sky atmospheric
emissivity, which was based on the analytic solution of the Schwarzschild's equation
for a standard atmospheric lapse rates of temperature and water vapor. Prata (1996)
found that the precipitable water content ($w$) was much better to represent the
effective emissivity of the atmosphere than $e$, which was loosely based on radiative
transfer simulations. Dilley and O'Brien (1998) adopted this scheme but tuned
empirically their parameterization using an accurate radiative transfer model. Given
the fact that clear-sky DLR is impacted by water vapor and temperature profile
(especially the inversion layer) and diurnal variation of $T_a$, a new model with two
more coefficients considering these effects on DLR was developed (Dupont et al.,
2008a).

In the presence of clouds, the total effective emissivity of the sky is remarkably

modulated by clouds. The existing clear-sky parameterization should be modified
according to the cloud fraction (CF) and other cloud parameters. CF is generally used
to represent a fairly simple cloud modification under cloudy conditions. Many
equations with cloudiness correction have been developed and evaluated by the DLR
measurements across the world (Crawford and Duchon, 1999; Niemela et al., 2001).
CF is widely obtained from surface human observations (Iziomon et al., 2003) that is
subjective in nature. CF can also be derived from DSR (Crawford and Duchon, 1999)
and/or DLR measurements (Durr and Philipona, 2004). Moreover, DSR or DLR
measurements with very high temporal resolution (for example, 1-min) can also
provide cloud type information (Duchon and Malley, 1999), and thereby allowing to
consider the effects of cloud types on DLR (Orsini et al., 2002). This indicates that
1-min DSR and DLR measurements are beneficial to the DLR parameterization.

With an average altitude exceeding 4 km above the sea level (a.s.l.), Tibetan

Plateau (TP), the largest mountain area in the world, exerts a huge influence on
regional and global climate through mechanical and thermal forcings (Wu et al.,
2007). TP is the region with very high sensitivity to climate change. The most rapid
warming rate over the TP occurred in the latter half of the 20[th] century was likely



associated with relatively large DLR increase. Duan and Wu (2006) indicated that
increase in low level nocturnal cloud amount and thereby DLR can partly explain the
increase in the minimum temperature, despite decreases in total cloud amount during
the same period. By using observed sensitivity of DLR to change in specific humidity
for the Alps, Rangwala et al. (2009) suggested that increase in water vapor appeared
to be partly responsible for producing the large warming over the TP. Since the
coefficients of these empirical models and their performances showed spatiotemporal
variations, establishment of localized DLR parameterizations over the TP is of highly
significance. Given the importance of DLR to climate change, further studies on the
DLR parameterization as well as DLR sensitivity to atmospheric variables are
desirable, which would expected to improve our understanding of climate change over
the TP (Wang and Dickinson, 2013).

DLR measurements with high temporal resolution using high quality radiometer

over the TP are quite scarce. So it is not surprising that there have been very few
studies on DLR and its parameterization. Wang and Liang (2009) evaluated clear-sky
DLR parameterizations of Brunt (1932) and Brutsaert (1975) at 36 globally
distributed sites, in which DLR data at two TP stations were used. Yang et al. (2012)
used hourly DLR data at 6 stations to study the major characteristics of DLR and the
all-sky parameterization of Crawford and Duchon (1999) was assessed. More recently,
Zhu et al. (2017) evaluated 13 clear-sky and 10 all-sky DLR models based on hourly
DLR measurements at 5 automatic meteorological stations over the TP. Note that the
CG3 pyrgemeters (Kipp & Zonen), the second class radiometer according to the
International Organization for Standardization (ISO) classification, were used to
measure DLR in these previous studies. The parameterization would thus be impacted
by a large measurement uncertainty (roughly 10% according to the CG3 manual).
Clear-sky and CF were determined with relative low temporal resolution, for example,
subjectively by human observer every 3 or 6 hours, which would also impact the
parameterization. One would expect that these previous methods developed for daily
or longer-term averages were usually less accurate at shorter time intervals.

In order to further our understanding of DLR and DSR over the TP,





measurements of 1-min DSR and DLR at 3 stations over the TP using state-of-the-art
instruments have been performed in summer months since 2011. These data provide
us opportunity to evaluate clear-sky DLR models and quantitatively assess how cloud
properties impact DLR. This study makes progress in the following aspects. Clear-sky
discrimination and CF estimation are based on 1-min DSR and DLR measurements
that are objective in nature. Misclassification of cloudiness into cloud-free skies
would be minimized by adopting strict cloud-screening procedures based on not only
1-min DSR and DLR measurements but also coincident Lidar backscattering
measurements. Potential effects of cloud-base height (CBH) on overcast DLR are
investigated. Locally calibrated parameterizations of clear- and cloudy-sky DLRs are
finally achieved.

**2. Site, Instrument and Data**
Measurements of DLR and DSR are conducted 1~4 months at three stations
(Table 1), including Nagqu (NQ, 92.04°E, 31.29°N, 4.5 km a.s.l), Nyingchi(NC,
94.2°E, 29.4°N, 2.3 km a.s.l.) and Ali (AL, 80°E, 32.5°N, 4.3 km a.s.l.). DLR and
DSR were measured by CG4 and CM21 radiometers, respectively.   The sampling
rate is 1 Hz and the averages of the samples over 1-min intervals are used.
Simultaneous 1-min averages of $T_a$ and $e$ are taken from the automatic meteorological
stations. CG4 is designed for the DLR measurement with high reliability and accuracy
due to its specific material and unique construction. Window heating due to
absorption of solar radiation in the window material is the major error source of DLR
measurement, which is strongly suppressed by a unique construction conducting away
the absorbed heat very effectively. The shading and un-shading experiment of CG4
measurements show a window heating offset of less than 4 $W \cdot m^{-2}$(Meloni et al., 2012),
as a comparison, it can reach 25 $W \cdot m^{-2}$ for CG3 since it is always not shaded (Wang
and Dickerson, 2013). An installation of the CG4 on the Kipp & Zonen CV2
ventilation unit is able to prevent dew deposition on the window. The radiometers are
calibrated before and after field measurements through comparison to the reference
radiometers operated by the national metrological standards of meteorology that is




ultimately traceable to the World Infrared Standard Group.
A Micropulse Lidar (MPL-4B, Sigma Space Corporation, United States) was
installed site-by-site with radiometers. The Nd:YLF laser of the MPL produces an
output power of 12 μJ at 532 nm. The repletion rate is 2500 Hz. The vertical
resolution of the MPL data is 30 m and the integration time of the measurements is 30
s. The MPL backscattering profiles are used to identify the cloud boundaries and
derive the CBHs (He et al., 2013). The dataset contains about 700 hours of coincident
DLR, DSR, Lidar and meteorological measurements.

**3. Methods**
**3.1 Clear-sky discrimination**
Clear skies should be discriminated from cloudy conditions before performing
clear-sky DLR parametrization, which is achieved by the synthetical analysis of DSR,
DLR, and CBH from MPL. The term clear sky or cloud-free in this paper means a sky
without any condensed liquid or ice water for all classes of altitude.
Following the method initiated by Crwford and Duchon (1999), we calculate two
quantities reflecting DSR magnitude and variability based on 1-min observed DSR
($DSR_{obs}$) and calculated clear-sky DSR ($DSR_{cal}$) values. $DSR_{cal}$ is calculated by the
model C of Iqbal (1983) in which direct and diffuse components of DSR on a
horizontal surface are parametrized separately. Direct DSR is first calculated by
multiplying transmittance due to Rayleigh scattering, aerosol attenuation and
absorption by water vapor, ozone and the uniformly mixed gases. Diffuse DSR is
estimated as the sum of the Rayleigh and aerosol scattering as well as the multiple
reflected irradiance between surface and atmosphere. The terrain reflection is
estimated according to Dozier and Frew (1990). The precipitable water is calculated
from $e$ according to a linear relationship that was developed based on collocated $e$ and
radiosonde (in AL) or GPS (NQ and NC) -based precipitable water measurements .
Climatological value of aerosol optical depth and single scattering albedo are from the
reference (Che et al., 2015). Mean surface albedo values of 0.22 at NQ, 0.18 at NC,
and 0.25 at AL were from Liang et al. (2012).




1-min $DSR_{cal}$ are first scaled to a constant value of 1400 $W \cdot m^{-2}$, which is used to
normalize the $DSR_{obs}$ by multiplying the same set of scale factors (Duchon and
Malley, 1999; Long and Ackerman, 2000; Orsini et al., 2002). The mean and standard
deviation of the scaled $DSR_{obs}$ in a 21-min moving window (±10-min) centered on the
time of interest are then calculated to discriminate clear-sky. Selection of the width of
21-min is empirical but a consequence of having a reasonable time span for
estimating the mean and variance (Duchon and Malley, 1999). Clear-sky DSR should
satisfy the followed three requirements: 1) ratio of $DSR_{obs}$ to $DSR_{cal}$ is within 0.95 to
1.05, 2) difference between scaled $DSR_{obs}$ and $DSR_{cal}$ is less than 20 $W \cdot m^{-2}$, and 3)
standard deviation of scaled $DSR_{obs}$ is less than 20 $W \cdot m^{-2}$. Temporal variability of
DLR is also used to separate cloudy sky from cloud-free situations. Based on analysis
of the standard deviation of scaled DLR (scaled to 500 $W \cdot m^{-2}$) for a ±10-min period,
clear-sky periods are detected if the standard deviation is less than 5 $W \cdot m^{-2}$. Given the
fact that DSR and DLR experience difficulties in detecting clouds in the portion of the
sky far away from the sun (Duchon and Malley, 1999) or high-altitude cirrus clouds
(Dupont et al., 2008b), coincident MPL backscatter measurements are used to strictly
select clear-sky samples. We can be sure that there is a cloud element somewhere in
the sky when the MPL identifies a cloud, we require that no clouds are detected by the
MPL within the ±10-min period, otherwise it is defined as cloudy condition.
These two different methods are complementary to each other to some extent
(Dupont et al., 2008b), one would expect that a combined analysis of both passive and
active remote sensing instruments can precisely detect clear sky periods. We hence
use the following strategy to select clear-sky samples. If DSR, DLR and MPL
measurements at the time of interest synchronously satisfy specified clear-sky
conditions, the sample is thought to be taken under unambiguously cloud-free
condition; on the contrary, the measurement are made under unambiguously cloudy
condition if all of these three methods suggests to be cloudy. Our following clear-sky
and cloudy DLR parameterizations are respectively based on measurements under
unambiguously cloud-free and cloudy conditions. A total of 8195-minutes clear-sky
samples and 69318-minutes cloudy-sky samples are used in the analysis.



230   Fig. 1 shows how our method determines clear-sky conditions. $DSR_{obs}$ presents a

231  smooth temporal variation from sunrise to about 14:00, August 19, 2016 (LST),

232  being consistent with $DSR_{clr}$. Similarly, DLR also varies very smoothly during this

233  period and standard deviations of 21-min DLRs are generally less than 5 W m$^{-2}$.

234  Both facts suggest that the sky is sunny and cloudless. This inference is supported by

235  MPL backscatter measurements that do not detect any clouds overhead. Contrarily,

236  an abruptly changes of 1-min $DSR_{obs}$ and DLR are evident and we can see $DSR_{obs}$

237  occasionally exceeds the expected $DSR_{clr}$, indicating occurrence of thin or fair

238  weather cumuli clouds. MPL detect a persistent cloud layer at 3 km above ground

239  during 14:00-17:00 LST, which agrees with DSR and DLR measurements very well.

240  Two-layer clouds are observed by MPL until to sunset, which is accompanied by

241  highly variation of observed DSR and DLR.

**3.2 Cloud fraction estimation**

244   Given synoptic cloud observations are very limited and temporally sparse, various

245  parameterizations using DSR or DLR data have been developed to estimate the cloud

246  fraction (CF) or called cloud modulate factor (CMF) (e.g., Deardorff, 1978; Marty

247  and Philipona, 2000; Durr and Philipona, 2004; Long et al., 2006; 2008). Because of

248  the good agreement between clear-sky $DSR_{obs}$ and $DSR_{cal}$ calculated by the Iqbal C

249  calculations (Iqbal, 1983), with mean bias of 1.7 Wm$^{-2}$ and root mean square error

250  (RMSE) of 10.7 Wm$^{-2}$, we use Deardorff 's method to calculate CF from $DSR_{obs}$ and

251  $DSR_{cal}$. The method is based on a fairly simple cloud modification to DSR as follows.

252    $$CF = 1 - \frac{DSR_{obs}}{DSR_{cal}} \qquad (1)$$

253   To avoid the error caused by abrupt DSR variation, 21-min DSR samples rather

254  than its instantaneous measurements are used to calculate CF here.


**4 Results**

257  4.1 Clear-sky DLR parameterization evaluation and localization

258   Eleven clear-sky DLR ($DLR_{clr}$) parameterizations (Table 2) are evaluated based



on 1-min DLR measurements under unambiguously cloud-free conditions. To
compare the performance of these 11 models, RMSE and the coefficient of
determination ($R^2$) are shown by a Taylor diagram in Fig. 2(a). The Brutsaert (1975);
Konzelmann (1994); Dilley and O'Brien (1998) and Prata (1995) models show
relatively smaller RMSE (generally < 15 W·m$^{-2}$) and larger R$^2$ (>0.95). One possible
reason is that those parameterizations were developed in cool and dry areas (like
England in Brutsaert (1975), Greenland in Konzelmann (1994) and Australian desert
in Prata (1995). The climate in those areas is likely similar to that in TP, so one would
expect the coefficients in those parameterizations are also suitable in TP. The higher
RMSE (>37 W m$^{-2}$) and the lower R$^2$ (~0.7) for Swinbank (1963) and Idso and
Jackson (1969). Both used $T$ as the sole parameter. The essential point was that the
screen temperature is a better indicator of the mass of radiatively active water vapor
than the surface vapor pressure. However, previous studies have suggested that these
methods would produce substantially large RMSE (>37.5 W·m$^{-2}$) and low R$^2$ (<0.75)
(Duarte et al., 2006). The reason is that the atmospheric effective emissivity is more
sensitive to the water vapor profile than the mass of radiatively active water vapor
when the surface layer is dry compared to the whole column (Dupon et al., 2008).
Furthermore, DLR$_{clr}$ is more much sensitive to variation of water vapor content over
the TP than humid environment. Careful consideration of water vapor effect on DLR
is obviously required over the TP.
The coefficients in eleven parameterizations (Table 2) were originally calibrated
and determined in different geographical locations; therefore, they may not be the
optimal values for the usage in the TP. Thus we take use of 1-min clear-sky DLR
samples to locally calibrate the parameters of these parametrizations. We used k-fold
cross-validation method to determine the local parameters. This method has two main
advantages:ⅰ)less error rate because it repeatedly fits the statistical learning method
using training data sets,. ⅱ)decreasing the error rate by using random
training/validation data sets for multiple times (James et al., 2013). Here, all data was
randomly divided into 10 groups of approximately equal size, the coefficients are
computed by using 9 groups as training set, and the remained one as validation. This





procedure is repeated 10 times to get the representational value (with the lowest test
error) of coefficients in different parameterizations.
The non-linear least-squares fitting of the $DLR_{clr}$ parameterizations (Table 2)
resulted in the coefficient values in Table 3. For each fitted parameterization, we
calculated RMSE and $R^2$ and the results are shown in Fig. 2(b). When using the
parameterizations with the locally fitted parameters (Fig. 2(b)), the accuracy of the
parameterization relative to the published values is substantially improved. Most
RMSEs are less than 10 W·m$^{-2}$ except the parameterization proposed by Swinbank
(1963) and Idso and Jackson (1969) that still produced the worst results (with $R^2$ of
0.71 and RMSE of 15 W·m$^{-2}$) even the parameters are locally calibrated. The Dilley
and O'Brien's parameterization, which was initially developed by considering the
adaptation of climatological diversities, is expected to be able to fit the measurements
in tropical, mid-latitude and Polar Regions. This expectation is verified by its wide
deployment in $DLR_{clr}$ estimations in different climate regimes and altitude levels, for
example, in the tropical lowland (eastern Pará state, Brazil) and the mild mountain
area (Boulder, the United States) (Marthews et al., 2012; Li et al., 2017). The present
study also confirmed that Dilley and O'Brien is the best clear-sky parameterization
over the TP. This parameterization was also proved to be the most reliable estimates
of $DLR_{clr}$ in the TP (Zhu et al., 2017). The locally calibrated equation is as follows.

$$DLR_{clr} = -2.53 + 158.10 \times \left(\frac{T}{273.16}\right)^6 + 106.40 \times \left(\frac{46.50 \times \frac{e}{T}}{2.50}\right)^{\frac{1}{2}} \qquad (2)$$

The RMSE and $R^2$ of Eq.(2) are ~3.8 W·m$^{-2}$ and > 0.98 respectively, which are
substantially lower than those in previous studies in the TP, for example, the RMSE
was 9.5 W·m$^{-2}$ in Zhu et al. (2017). Note that the parameters here differ quite a lot
from those in the reference (Zhu et al., 2017) that is shown in Eq. (3).

$$DLR_{clr} = 30.00 + 157.00 \times \left(\frac{T}{273.16}\right)^6 + 97.93 \times \left(\frac{46.50 \times \frac{e}{T}}{2.50}\right)^{\frac{1}{2}} \qquad (3)$$

Fig.3 shows the comparison of instantaneous clear-sky DLR measurements as a
function of calculations by Eq. (2) and by Eq. (3). It is seen that measurements are in
good agreement with calculations of Eq. (2), as shown by an overwhelmingly large
number of data points falling along or overlap the 1:1 line. By contrast, clear-sky



DLR is always overestimated by Eq. (3). Note that Eq. (3) was derived from 1-hour
DLR measurements, which was discriminated to be taken under clear-sky or cloudy
conditions based on human observation at even lower resolution (every 3-6 hours).
Both factors are likely to introduce potential cloud contamination on clear-sky
discrimination due to rapid variations of cloud. The presence of clouds would lead to
a larger DLR value relative to that in clear sky, which is most likely cause for the
overestimation of Eq. (3). Significant impacts on the monthly and yearly radiation
budget of the same magnitude are not avoided as a result of persisting overestimation
of DLR by Eq. (3).

**4.2 Parameterization of cloudy-sky DLR**
The parameterizations of cloudy-sky DLR (DLR$_{cld}$) are based on estimated
DLR$_{clr}$ coupled with the effect of cloudiness or cloud emissivity, which depends
primarily on CF, and some other cloud parameters, like CBH and cloud type (Arking,
1990; Viúdez-Mora et al., 2014). Four parameterizations (Table 4), which modifies
the bulk emissivity depending on CF, are assessed and locally calibrated in this
section.
DLR$_{clr}$ is estimated according to Eq. (2) with the locally fitted coefficients. The
fitted values of the coefficients (using k-Fold Cross-Validation) of the four
parameterizations are presented in Table 5, and the RMSE and $R^2$ of original and
locally fitted parameterizations in TP are presented in Fig. 4.
Relative to that under clear-sky conditions, cloudy parameterizations using the
given parameters produced larger RMSE (generally exceeding 35 W·m$^{-2}$) except that
developed by Jacobs (1978) (RMSE of 18 W·m$^{-2}$). $R^2$ was generally smaller than 0.9.
RMSE decreased significantly in Maykut and Church (1973) and Sugita and Brutsaert
(1993) as locally calibrated parameters were used. Relative smaller and almost no
RMSE improvements were found for the methods developed by Konzelmann (1994)
and Jacobs (1978).
Eq. (4) shows the best cloudy-sky parameterization over TP by combining the
clear-sky parameterization of Dilley and O'Brien (1998) with the cloud modulation





348 correction scheme of Jacobs (1978).

349 $$DLR_{cld} = (1 + 0.23 \times CF) \ \times \ (59.38 + 113.70 \times \left(\frac{T}{273.16}\right)^6 + 96.96 \times \left(\frac{46.50 \times \frac{e}{T}}{2.50}\right)^{\frac{1}{2}}) \qquad (4)$$

350 The RMSE and $R^2$ are ~18 $W \cdot m^{-2}$ and ~0.89. RMSE here is close to 15 $W\ m^{-2}$

351 obtained at different altitudes in Swiss (Gubler et al., 2012), and slightly lower than

352 23 $W\ m^{-2}$ in mountain area in Germany (Iziomon et al., 2003). Comparing to previous

353 studies over the TP (RMSE of 22 $W\ m^{-2}$ in Zhu et al., 2017), our cloudy model also

354 produces better results.

356 **4.3 Effect of CBH on DLR under Overcast Conditions**

357 Since clouds behave approximately as a blackbody, the most relevant cloud

358 parameter (besides CF) to DLR under overcast skies ($DLR_{ovc}$) is the temperature of its

359 lower boundary (CBH). Radiative transfer model simulation has suggested that CBH

360 under overcast conditions is an important modulator for the DLR. The cloud radiation

361 effect (CRE), the difference between $DLR_{obs}$ and $DLR_{clr}$, decreases with increasing

362 CBH at a rate of 4~12 $W \cdot m^{-2}$ that depends on climate profiles (Viúdez-Mora et al.,

363 2014). This indicated that cloudy DLR parameterization can be improved if CBH

364 effect is considered.

365 The statistical relationship between CRE and CBH under overcast conditions in

366 the TP is presented in Fig. 5, a box plot of CBH versus CRE. The peak and median

367 values of CRE decrease with the increase of CBH. With the increase of CBH, the

368 variation range of the CRE rises, ranging from 25 to 50 $W \cdot m^{-2}$, as a result of the

369 specific meteorological and cloud conditions. Compared to that at Girona, Spain, a

370 low altitude mid-latitude site (Viúdez-Mora, et al., 2014), CRE in the TP is generally

371 lower by 5~10 $W \cdot m^{-2}$. This is likely associated with the fact that clouds in the TP with

372 the same CBH as that in Girona have relatively lower temperature, thereby producing

373 lower radiative effect on DLR. It is interesting that the decreasing tendency of CRE

374 with CBH is apparent. CRE is about 70 $W \cdot m^{-2}$ for clouds < 1 km and decreases to ~40

375 $W \cdot m^{-2}$ for clouds at 3~4 km in the TP. The decreasing rate of CRE with CBH is

376 estimated to be -9.8 $W \cdot m^{-2} km^{-1}$ in the TP that is within the model simulations



(Viúdez-Mora et al. 2014).
To consider CBH effect under overcast conditions, we introduced a modified
parameterization similar as that in Viúdez-Mora et al. (2014).
$DLR_{ovc} = 1.23 \times DLR_{clr} \times (1.01 - 0.06 \times \text{CBH})$          (5)
The bias and RMSE of Eq.(5) between measurements and calculations is 1.3
W·m$^{-2}$ and 16.5 W·m$^{-2}$, respectively, which are significantly lower than that of Eq.(4)
(10.3 W m$^{-2}$ and 21.4 W m$^{-2}$) in overcast conditions. This result indicates a remarkable
improvement in the estimation of DLR under overcast conditions by introducing CBH
to the DLR parameterization.

**5 Discussion and conclusions**
The parameterization of clear-sky DLR requires a well-defined distinction
between clear-sky and cloudy-sky situations that commonly depends on human cloud
observations 4~6 times each day. Human observations are subjective in nature and
have a very limited temporal resolution that obviously cannot capture dramatic
variations of clouds. Furthermore, synoptic(human cloud observations show the
tendency to stronger weight the horizon that DLR is not highly sensitive (Marty and
Philipona). Therefore, parameterization of clear-sky DLR based on synoptic sky
observations is hence very likely biased as a consequence of improper selection of
clear-sky measurements. This issue should be considered cautiously because it is
essential to precisely quantify aerosol and cloud radiative effects that rely on precise
identification of cloud free references (Dupont et al., 2008b).
Using 1-min DSR and DLR at 3 stations over the TP, DLR parameterizations are
evaluated and localized parameterizations have been developed. Potential CBH effect
on overcast DLR is experimentally determined. Major conclusions are as follows.
Among 11 clear-sky DLR parameterizations tested in this study, these two
methods using only atmospheric temperature largely deviated from other
parameterizations. DLR estimation can be improved by localization of these
parameterizations. The best method suitable for the TP is the parameterization



developed by Dilley and O'Brien (1997). The locally calibrated Dilley and O'Brien
model can produce clear-sky DLR with a RMSE of 3.8 W·m$^{-2}$.
Overcast DLR is highly sensitive to CBH. The parameterization in this case can
be substantially improved by consideration of CBH effect. The bias between model
calculations and measurements decreases from 10.3 W m$^{-2}$ to 1.3 W m$^{-2}$ when CBH
effect is introduced
A broadly representative of existing DLR parameterizations with good
performance was assessed over the TP, while this did not imply that our sample of
techniques was either exhaustive or optimal in all applications. We only focused on
daytime DLR parameterization in TP since DSR is used in the cloud-screening
method. Given a significant role of DLR played in the surface energy budget during
nighttime, it is highly desirable to perform further study on the nighttime DLR
parametrization in future. These results are based on summer DLR measurements in
TP, so the conclusions here need to be tested in other seasons, especially in winter
when DLR has been observed to increase in the TP (Rangwala et al., 2009). These
further study would shed new light on how DLR is related to temperature and water
vapor and why DLR has changed in the TP.

Author contributions. XD and XA designed the experiments and MQ carried them out.
MQ and JQ prepared the manuscript with contributions from all co-authors.
Competing interests. The authors declare that they have no conflict of interest.
Data availability. The data can be obtained from the corresponding author upon
request.





Acknowledgements: This work was supported by the Strategic Priority Research
Program of Chinese Academy of Sciences (XDA17010101), the National Key R&D
Program of China (2017YFA0603504), the National Natural Science Foundation of
China (91537213 and 91637107), the Special Fund for Meteorological Research in
the Public Interest (GYHY201106023), and the Science and Technological Innovation
Team Project of Chinese Academy of Meteorological Science (2013Z005). We greatly
appreciate Dr. Q. He for providing the MPL Lidar measurement images and derived
CBH data.









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






Table 1: Description of station and measurement (magnitude and variability) in the

Tibetan Plateau

| Site | Altitude (m) | Period | T (°C) | $e$ (hPa) | DLR (W·m$^{-2}$) | Data Points |
|------|----------|--------|--------|---------|-----------------|-------------|
| NQ | 4507 | 2011.7.20-2011.8.26 | 9.4±8 | 7.4±5 | 242.75±40 | 52980 |
| NC | 2290 | 2014.6.7-2014.7.31 | 16.8±10 | 13.4±4 | 368.25±40 | 69609 |
| AL | 4279 | 2016.5.27-2016.9.22 | 7.8±4 | 4.8±4 | 253.11±50 | 86596 |





**Table 2.** 11 clear-sky DLR parameterizations and associated specific conditions

| Reference | Clear-Sky Parameterization | Conditions |
|---|---|---|
| Angstrom (1915) | $DLR_{clr} = \{0.83 - 0.18 \times 10^{-0.067e}\}\sigma T^4$ | Alt.: 1650, 3500m a.s.l<br>T: 10~30°C<br>$e$: 4~17hPa |
| Brunt (1932) | $DLR_{clr} = (0.52 + 0.065\sqrt{e})\sigma T^4$ | Alt.: 6, 1650, 3500m a.s.l<br>T: -4~30°C<br>$e$: 2.5~16hPa |
| Swinbank (1963) | $DLR_{clr} = 5.31 \times 10^{-13}T^6$ | Alt: 2m a.s.l<br>T: 8~29°C<br>$e$: 8~30hPa |
| Idso and Jackson (1969) | $DLR_{clr} = (1 - 0.261$ $\cdot \exp(-0.000777$ $\times (273 - T)^2))\sigma T^4$ | Alt.: 3, 331m a.s.l<br>T: -45~45°C |
| Brutsaert (1975) | $DLR_{clr} = 1.24\left(\frac{e}{T}\right)^{\frac{1}{7}}\sigma T^4$ | Alt.: 6, 1650, 3500m a.s.l<br>T: -4~30°C<br>$e$: 2.5~-16hPa |
| Satterlund (1979) | $DLR_{clr} = 1.08\left(1 - \exp\left(-e^{\frac{T}{2016}}\right)\right)\sigma T^4$ | Alt.: 594m a.s.l<br>T: -37~36°C<br>$e$: 0~18hPa |
| Idso (1981) | $DLR_{clr} = \left(0.7 + 5.95 \times 10^{-5} \times e\right.$ $\left. \times \exp\left(\frac{1500}{T}\right)\right)\sigma T^4$ | Alt.: 331m a.s.l<br>T: -15~5°C<br>$e$: 2~6hPa |
| Konzelmann (1994) | $DLR_{clr} = \left(0.23 + 0.443\left(\frac{e}{T}\right)^{\frac{1}{8}}\right)\sigma T^4$ | Alt.: 340~3230m a.s.l<br>T: -16~6°C<br>$e$: 1.5~5.5hPa |
| Prata (1995) | $DLR_{clr} = (1-(1+46.5\frac{e}{T}) \times \exp(-(1.2+3\times46.5$ $\frac{e}{T})^{0.5}))\,\sigma T^4$ | Not specified |
| Dilley and O'Brien (1998) | $DLR_{clr} = 59.38 + 113.7\left(\frac{T}{273.16}\right)^6 +$ $96.96\sqrt{46.5\frac{e}{T}/2.5}$ | Not specified |
| Iziomon (2001) | $DLR_{clr} = \left(1 - 0.43\exp\left(-\frac{11.5e}{T}\right)\right)\sigma T^4$ | Alt.: 1489m a.s.l<br>$\bar{T}$=4.4°C<br>$\bar{e}$ =7.4hPa |

*Where $e$ is screen-level water vapor pressure in hPa and T represents surface temperature in K





**Table 3.** Locally fitted clear-sky DLR parameterizations in TP

| Reference | Locally fitted Clear-Sky Parameterization |
|---|---|
| Angstrom(1915) | $DLR_{clr} = \{0.8 - 0.19 \times 10^{-0.068e}\}\sigma T^4$ |
| Brunt(1932) | $DLR_{clr} = (0.56 + 0.07\sqrt{e})\sigma T^4$ |
| Swinbank(1963) | $DLR_{clr} = 4.7 \times 10^{-13}T^6$ |
| Idso & Jackson(1969) | $DLR_{clr} = (1 - 0.36 \cdot \exp(-0.00065 \times (273 - T)^2))\sigma T^4$ |
| Brutsaert(1975) | $DLR_{clr} = 1.03 \left(\dfrac{e}{T}\right)^{0.09} \sigma T^4$ |
| Satterlun (1979) | $DLR_{clr} = \left(1 - \exp\left(-e^{\frac{T}{2016}}\right)\right)\sigma T^4$ |
| Idso(1981) | $DLR_{clr} = \left(0.63 + 7.5 \times 10^{-5} \times e \times \exp\left(\dfrac{1500}{T}\right)\right)\sigma T^4$ |
| Konzelmann(1994) | $DLR_{clr} = \left(0.23 + 0.45 \left(\dfrac{e}{T}\right)^{0.13}\right)\sigma T^4$ |
| Prata(1995) | $DLR_{clr} = (1-(1+46.5\tfrac{e}{T}) \times \exp(-(1+3\times46.5\tfrac{e}{T})^{0.5}))\,\sigma T^4$ |
| Dilley and O'Brien(1998) | $DLR_{clr} = -2.54 + 158.1\left(\dfrac{T}{273.16}\right)^6 + 106.4\sqrt{46.5\tfrac{e}{T}/2.5}$ |
| Iziomon(2001) | $DLR_{clr} = \left(1 - 0.38\exp\left(-\dfrac{14.52e}{T}\right)\right)\sigma T^4$ |




**Table 4.** 4 Cloudy-sky DLR Parameterizations in the references

| Reference | Cloudy-Sky Parameterization |
|---|---|
| Maykut and Church, 1973 | $DLR_{cld} = (0.7855 + 0.000312CF^{2.75})\sigma T^4$ |
| Jacobs, 1978 | $DLR_{cld} = (1 + 0.26CF)DLR_{clr}$ |
| Sugita and Brutsaert, 1993 | $DLR_{cld} = (1 + 0.0496CF^{2.45})\ DLR_{clr}$ |
| Konzelmann, 1994 | $DLR_{cld} = (1 - CF^4)DLR_{clr} + 0.954CF^4\sigma T^4$ |





**Table 5.** Locally fitted cloudy-sky DLR parameterizations in TP

| Reference | Locally fitted Cloudy-Sky Parameterization |
|---|---|
| Maykut and Church, 1973 | $DLR_{cld} = (0.85 + 0.01CF^3)\sigma T^4$ |
| Jacobs, 1978 | $DLR_{cld} = (1 + 0.23CF)DLR_{clr}$ |
| Sugita and Brutsaert, 1993 | $DLR_{cld} = (1 + 0.2CF^{1.3})\ DLR_{clr}$ |
| Konzelmann, 1994 | $DLR_{cld} = (1 - CF^{3.5})DLR_{clr} + CF^{3.5}\sigma T^4$ |







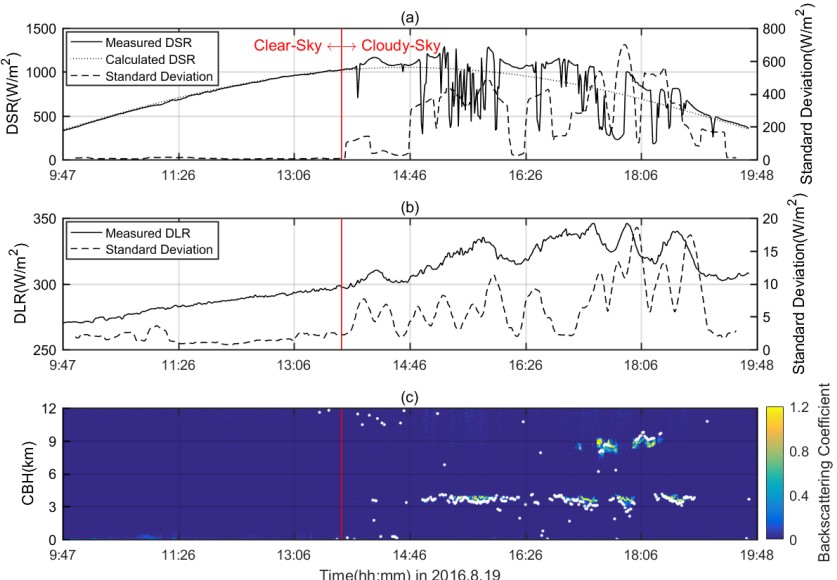


Fig. 1. Time series of one-day sample on 2016.8.19 transited from clear-skies to cloudy-skies: (a)
measured (black line) and calculated (dotted black line) downward shortwave radiation and its
21-min standard deviation (grey line), (b) measured downward longwave radiation and 21-min
standard deviation and (c) MPL backscattering coefficient (color bar) and the cloud base height
(white dots).

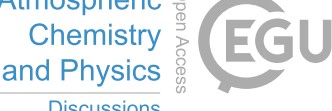

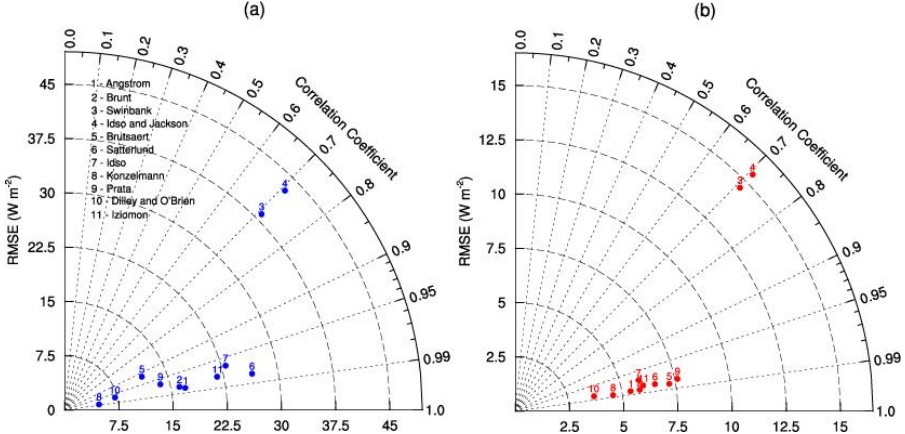

Fig. 2. RMSE and $R^2$ for the clear-sky DLR parameterizations using original (a) and
locally calibrated (b) coefficient values.






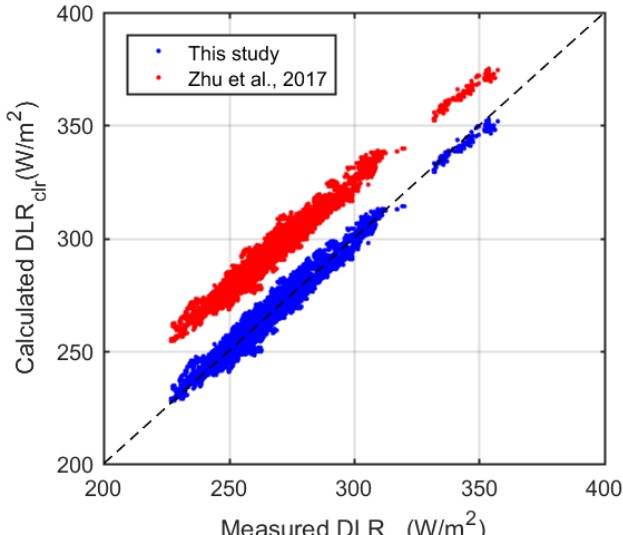


Fig. 3. Scatter plots of instantaneous clear-sky DLR data from measurements as a
function of calculations by this study (blue dots) and by Zhu et al. (2017) (red
dots).The dash black line is the 1:1 line.



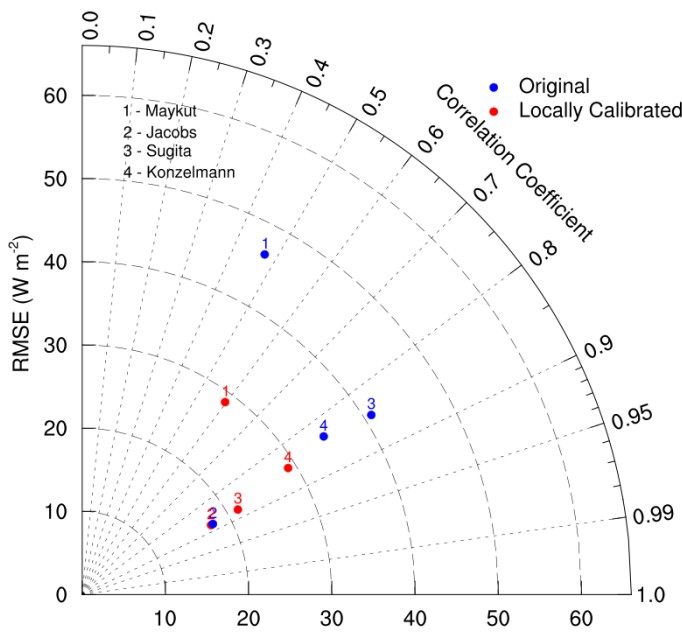

Fig. 4. RMSE and $R^2$ for the cloudy-sky DLR ($DLR_{cld}$) parameterizations using the
original (blue) and locally calibrated (red) coefficient values.



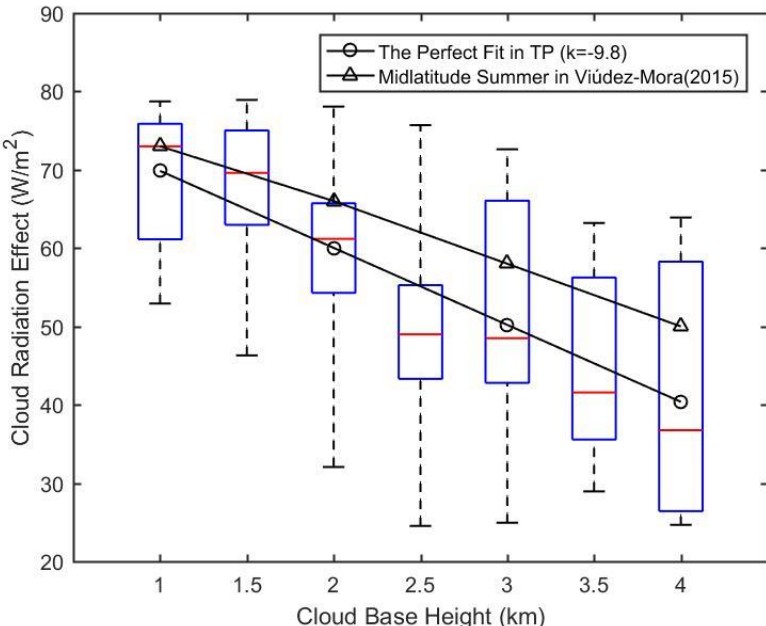


Fig. 5. Scatter plot of cloud radiative effect against MPL derived cloud base height are
represented by box plot (the blue box indicates the 25th and 75th percentiles, the
whiskers indicate 5th and 95th percentiles, the red middle line is the median). The
black circles line and the black triangles is mean values of cloud radiative effect over
TP in this study and in Girona, Spain (Viúdez-Mora et al., 2014).