# Peer review of "A revisit of parametrization of summer downward longwave"

_Atmospheric Chemistry and Physics, 2019_

## Referee Comment (RC1) · Anonymous Referee #1 · 13 Jun 2019

Review of acp-2019-397: "A revisit of parametrization of summer downward longwave radiation over the Tibetan Plateau from high temporal resolution measurements" by Liu et al. This paper uses high temporal resolution measurements to evaluate the existing downward longwave radiation (DLR) parameterizations under clear-sky, cloudy and overcast conditions at the Tibetan Plateau (TP). The authors have done a good job in the literature review and the data is valuable. The careful discrimination of clear-sky is also meaningful. However, this manuscript does not report significant advances nor novel aspects of experimental and theoretical methods and techniques. The major

conclusions, such as the best DLR parameterization scheme that is suitable for TP have been reached by other researchers, such as Zhu et al 2017, as mentioned in the paper. The detailed comments are listed below. Major Comments: 1. Improved DLR estimation: In the abstract, as well as in Ln 353, the authors state that the DLR estimation is notably improved after local calibration. I think this statement is misleading. The authors use existing parameterizations to fit the data measured at TP. And for sure, the same fitting equation but with different coefficients would give better results compared with the literature parameterization that uses coefficients derived from measurements conducted at different places or at different conditions. 2. Different parameters with Zhu et al (2017): Were the datasets of DLR, e, and T used in this manuscript measured at the same time and same sites compared with Zhu et al 2017? Otherwise, it might not be appropriate to say the difference is caused by cloud contamination (Ln 321-324). The difference can also be caused by different DLR magnitudes. Minor Comments: 1. Ln 27: 'highly sensitive'—-'high sensitivity' 2. Ln 34: 'by making maximal use of'—-'by making the maximal use of' 3. Ln 63: What is the '2-sigma uncertainty of DLR measurement'? 4. Ln 120: 'would expected'—-'would be expected' 5. Ln 141: 'since 2011'——Did you mean in 2011? 6. Ln 153: What the specific measurement periods for the three stations are? 7. Ln 156: The authors give detailed description of CG4. How about CM21? 8. Ln 269: '. Both used T...'—-'both used T...' 9. Ln 282: Add reference for k-fold cross-validation method. 10. Ln 369: Can you give examples of the 'specific meteorological and cloud conditions'? 11. Ln 371-372: What is the supporting evidence for the 'fact that that clouds in the TP with the same CBH as that in Girona have relatively lower temperature'? 12. Equations in this manuscript should be followed by definitions of each parameter and corresponding units.

---

## Referee Comment (RC2) · Anonymous Referee #2 · 23 Jun 2019

The parameterizations used to calculate DLR are pretty outdated, as stated in the introduction and other places in this study. One question is that the empirical parameterizations, for example those used in this study, are strongly dependent on locations and time and thus might be suitable for specific locations and seasons but not for others. As the authors stated in introduction 'Understanding of complex spatiotemporal variation of DLR and its implication is essential for improving weather prediction, climate simulation as well as water cycling modeling'. The empirical parameterizations are apparently not able to obtain complex spatialtemporal variations of radiation flux. Actually,

an accurate radiation transfer model would be a better choice to calculate radiation flux. Cloud optical properties, especially cloud optical depth is critical to modulate radiation flux, which unfortunately has not taken into account in this study. Also, a simple way is used to calculate cloud fraction (equation 1) in the manusctipt, so it is necessary to evaluate the calculated values with the observed ones at the meteorological site over TP.

Minor comments: The references cited in introduction are pretty outdated. Are there any updated references on such kind of studies? Line 185-186: it is better to give an equation on how to calculate DSR. Line 189-192: how to deal with aerosol (concentrations, vertical profile, scattering and absorption, etc.) in your calculations? some details are better provided. Line 193-194: 'The terrain reflection is estimated according to Dozier and Frew (1990)', again please give some descriptions on how to estimate surface albedo. Line 197-199: give some description on why use these values as surface albedo, are they from surface measurements? Line 200: why scaled DSR to 1400W m-2, DSR is net downward shortwave radiation, rather than total solar radiation.

In addition, the paper would be greatly enhanced with additional proof reading to improve the quality of the written English.

---

## Author Comment (AC1) · 3 Aug 2019

General remarks: Review of acp-2019-397: "A revisit of parametrization of summer downward longwave radiation over the Tibetan Plateau from high temporal resolution measurements" by Liu et al. This paper uses high temporal resolution measurements to evaluate the existing downward longwave radiation (DLR) parameterizations under clear-sky, cloudy and overcast conditions at the Tibetan Plateau (TP). The authors have done a good job in the literature review and the data is valuable. The careful discrimination of clear sky is also meaningful. However, this manuscript does not report

significant advances nor novel aspects of experimental and theoretical methods and techniques. The major conclusions, such as the best DLR parameterization scheme that is suitable for TP have been reached by other researchers, such as Zhu et al 2017, as mentioned in the paper. Reply: We greatly appreciate the reviewer's efforts on reviewing our manuscript. Yes, DLR parameterization has been widely studied across the world, even including DLR in the TP. It should be noted that most parameterizations are based on hourly measurements of DLR and meteorological variables, such as Zhu et al., (2017) and Wang and Liang, (2009). Our major point is that clear-sky DLR parameterization may be seriously impacted by clear-sky data samples that are very likely contaminated by cloud residuals if human observations of cloud or hourly DLR measurements are used as the unique criteria in selecting data samples. Our result (Figure 3) clearly showed that clear-sky DLR in the previous studies was very likely overestimated by cloud residuals, which would significantly affect studies that take the clear-sky DLR estimation as their prior requirement, for example, cloud DLR forcing. Moreover, we studied the relationship between cloud base height and DLR that has never been investigated in the TP before. We consider these are our original contributions to our understanding of DLR parameterization in the TP. This research would be not possible if a comprehensive measurement project had not been performed. As one of important parts of a cooperated field campaign, the state-of-the-art pyranometer and pyrgeometer with ventilation and heating system are used to respectively measure downward shortwave and longwave radiation with 1-minute resolution, in addition, Lidar measurements provide much more information about clouds than before. To our best knowledge, installation of radiometers and Lidar site by site has never been performed, furthermore, 1-minute measurements are very rarely reported in the TP. These should be our novel aspects of experimental method, which indeed favors for our DLR parameterization study.

The detailed comments are listed below.

Major Comments: 1. Improved DLR estimation: In the abstract, as well as in Ln 353,

the authors state that the DLR estimation is notably improved after local calibration. I think this statement is misleading. The authors use existing parameterizations to fit the data measured at TP. And for sure, the same fitting equation but with different coefficients would give better results compared with the literature parameterization that uses coefficients derived from measurements conducted at different places or at different conditions. Reply: Many DLR parameterizations have been created based on local collocated DLR and meteorological data in the literatures. Application of these methods to every specific location generally includes two aspects. The first is to select the best parameterization formula that is most suitable for the local condition. The second is to derive local coefficients based on collocated DLR and meteorological observations. We tested a few widely used parameterizations and recommended one parameterization with the best performance that is able to improve the DLR estimation in the TP. We modified our manuscript according to these considerations as follows. Comparing to previous studies, DLR parameterizations here are shown be characterized by smaller root mean square error (RMSE) and higher coefficient of determination (R2).

2. Different parameters with Zhu et al (2017): Were the datasets of DLR, e, and T used in this manuscript measured at the same time and same sites compared with Zhu et al 2017? Otherwise, it might not be appropriate to say the difference is caused by cloud contamination (Ln 321-324). The difference can also be caused by different DLR magnitudes. Reply: Data of DLR, e, and T used in our study are not same in time or site as those used by Zhu et al. (2017). Hourly measurements are used by Zhu et al. (2017) but we use 1-mintue measurements. Our major point is that caution should be paid to the DLR parameterizations based on hourly or daily DLR and meteorological measurements. Data used in Zhu et al. (2017) are not available to us. We only take the parameterization formula recommended by Zhu et al. (2017) to compare our clear-sky measurements and parameterization. Different DLR measurements may contribute to the difference in clear-sky parameterization, however, we tend to suggest that it is very likely contaminated by residual cloud contamination based on the following reasons. First, in Figure 3, mean DLR values from measurements, our parameterization and

Zhu et al. formula are 268.6±19.7 Wm-2, 268.7±19.4 W.m-2, and 295.0±18.4 W.m-2, respectively. The result from Zhu et al. exceeds the measurements by 25 W.m-2 (10%), that is much more than the expected uncertainty of the measurements (2.5% or 4 W m-2) (Stoffel, 2005). This implies that different measurements cannot explain this large systematic bias. Second, the method of clear–sky identification in Zhu et al (2017) based on the DLR observation (Marty and Philipona, 2000) has its potential shortcoming. This method had been further assessed by Sutter et al. (2004) who stated that "the thin high cloud" can be misclassified as clear sky. More important, comparison of cloudy DLR parameterizations between this study and Zhu et al. (2017) showed good agreement (Figure below). Therefore, we tend to think that cloud residuals should be the major contributor to the difference. We discuss this issue in the revised manuscript.

Minor Comments: 1. Ln 27: 'highly sensitive'—'high sensitivity' Reply: Done, thanks. 2. Ln 34: 'by making maximal use of'—'by making the maximal use of' Reply: We revised this sentence as follows Three independent methods are used to discriminate clear sky from clouds based on 1-minute downward shortwave, longwave radiation measurements as well as Lidar data. 3. Ln 63: What is the '2-sigma uncertainty of DLR measurement'? Reply: sigma here means standard deviation, if the distribution of uncertainty of measurements is taken be Gaussian, 2-sigma uncertainty means the uncertainty of 95.5% of measurements is within this range. 4. Ln 120: 'would expected'—'would be expected' Reply: Done, thanks. 5. Ln 141: 'since 2011'—– Did you mean in 2011? Reply: We use the same instruments in these 3 stations. The measurements are made in summer, 2011 in NQ, 2014 in NC, and 2016 in AL. 6. Ln 153: What the specific measurement periods for the three stations are? Reply: Information is presented in Table 1. We omit this information to keep the text concise. 7. Ln 156: The authors give detailed description of CG4. How about CM21? Reply: CM21 is a high performance research grade pyranometer. It uses the same detector as CM11 that is used by many studies, but introduction of individually optimized temperature compensation for CM21 makes it having much a smaller thermal offset than CM11. 8. Ln 269: '. Both used T: : :'—'both used T: : :' Reply: Done, thanks. 9. Ln 282:

Add reference for k-fold cross-validation method. Reply: Done, thanks. 10. Ln 369: Can you give examples of the 'specific meteorological and cloud conditions'? Reply: CRE variation increases from 25 to 50 Wâÿśm-2 as CBH increases because water vapor influence and its variation goes up. 11. Ln 371-372: What is the supporting evidence for the 'fact that that clouds in the TP with the same CBH as that in Girona have relatively lower temperature'? Reply: This is because the altitude of stations in the TP is much higher than that in Girona. We comment on this in the revised manuscript. 12. Equations in this manuscript should be followed by definitions of each parameter and corresponding units. Reply: Done, thanks.
* * *
[Figure]

[Figure]

**Fig. 1.** Comparison of cloudy DLR parameterizations between this study and Zhu et al. (2017)

---

## Author Comment (AC2) · 3 Aug 2019

General Remarks.

The parameterizations used to calculate DLR are pretty outdated, as stated in the introduction and other places in this study. One question is that the empirical parameterizations, for example those used in this study, are strongly dependent on locations and time and thus might be suitable for specific locations and seasons but not for others. As the authors stated in introduction 'Understanding of complex spatiotemporal variation of DLR and its implication is essential for improving weather prediction, climate simulation as well as water cycling modeling'. The empirical parameterizations are apparently not able to obtain complex spatial-temporal variations of radiation flux. Actually, an accurate radiation transfer model would be a better choice to calculate radiation flux. Cloud optical properties, especially cloud optical depth is critical to modulate radiation flux, which unfortunately has not taken into account in this study. Also, a simple way is used to calculate cloud fraction (equation 1) in the manuscript, so it is necessary to evaluate the calculated values with the observed ones at the meteorological site over TP.

Reply: We greatly appreciate the reviewer's opinions on our submission. We revised the manuscript according to these comments and suggestions.

Yes, not only DLR but also any radiation flux can be calculated from an accurate radiative transfer model if information about atmospheric radiatively active compositions is well known, but unfortunately, our knowledge of these radiatively active compositions are very limited under many circumstances. Regarding DLR estimation in specific, information about cloud amount, type, phase, height is more or less related to DLR, let it alone remarkable effects of water vapor content and its profile under clear sky condition on DLR. Much progress has been made on DLR derivation from satellite measurements, however, satellite remote sensing DLR products are still not free of large uncertainty (Zhou et al., 2007; Ahn et al., 2018), especially in the regions of elevated or complex terrain. As pointed out by the reviewer, the empirical parameterizations have limitation, but their advantages are also apparent. The method is simple but effective in the estimation of DLR, especially in regions with the parameterizations locally adjusted by high quality DLR measurements. Moreover, meteorological variables used for the DLR estimation are available across the world. These apparent advantages make this method is still widely used by the community and contribute to our understanding of the energy budget of the Earth's system (Wang et al., 2013).

Cloud optical depth is a key factor affecting DLR. COD is generally derived from satellite measurements, however, it should be noted that large uncertainty is still associated with satellite COD retrievals in the regions of elevated and complex terrain. The advantage of the DLR parameterization lies in that it adopt surface meteorological observations as the major inputs. Therefore, it is not common to adopt COD in the DLR parameterizations since COD data are generally not available.

Human cloud observation every 3 or 6 hours are available in meteorological stations before 2013, however, this observation protocol is stopped afterwards. Therefore, human cloud observations are very limited to collocate with our cloud derivations from 1-minute DSR measurements that prevents our attempt to compare

cloud cover from human observations and our estimations.

Zhou, Y., Kratz, D. P., Wilber, A. C., Gupta, S. K., & Cess, R. D., An improved algorithm for retrieving surface downwelling longwave radiation from satellite measurements. J. Geophys. Res., 112(D15), 2007.

Ahn, S. H. , Lee, K. T. , Rim, S. H. , Zo, I. S. , & Kim, B. Y., Surface downward longwave radiation retrieval algorithm for GEO-KOMPSAT-2A/AMI. Asia-Pacific J. Atmos. Sci., 54(2), 237-251, 2018.

Wang, K., and Dickinson, R. E.: Global atmospheric downward longwave radiation at the surface from ground-based observations, satellite retrievals, and re-analyses, Rev. Geophys., 51, 150-185, 10.1002/rog.20009, 2013

Minor comments:
1.  The references cited in introduction are pretty outdated. Are there any updated references on such kind of studies?

Reply: we appreciate reviewer's comment here, in the revised version, we update some references in the introduction.

2.  Line 185-186: it is better to give an equation on how to calculate DSR.

Reply:

$$DSR_{dir} = 1S_0 \tau_r \tau_W \tau_o \tau_a \tau_g$$

where $\tau_r, \tau_w$, $\tau_o$, $\tau_a$ and $\tau_g$ are transmittances due to Rayleigh scattering, water vapor absorption, ozone absorption, aerosol extinction and absorption by uniformly mixed gases $O_2$ and $CO_2$, respectively. Diffuse radiation is estimated as the sum of the Rayleigh scattered, the aerosol-scattered and the multiple reflected irradiance.

3.  Line 189-192: how to deal with aerosol (concentrations, vertical profile, scattering and absorption, etc.) in your calculations? Some details are better provided.

Reply: $DSR_{cal}$ calculation needs the aerosol parameters as follows: Angstrom exponent (α), the Angstrom turbidity (β), single-scattering albedo (ω). For α, and β in NC and AL, the data are from the monthly average of in-situ Cimel photometer measurements. The data in NQ are adopted the same value in AL because both site are at similar high altitude. For ω, we use the average value of 0.90 retrieved from CIE-318 observation in Lhasa (91.13, 29.67, 3663m). Aerosol vertical profile is not considered.

4.  Line 193-194: 'The terrain reflection is estimated according to Dozier and Frew (1990)', again please give some descriptions on how to estimate surface albedo.

Reply: We deleted the terrain reflection component since our measurements were made under conditions with no surrounding mountains around sites.

5.  Line 197-199: give some description on why use these values as surface albedo, are they from surface measurements?

Reply: Yes, these values are from the surface measurements.
For NQ and AL, the surface albedo value are 0.25 and 0.22, which are derived from the reference (Liang et al., 2012)

Liang H., Zhang R., Liu J., Sun Z., and Cheng X., Estimation of Hourly Solar Radiation at the Surface under Cloudless Conditions on the Tibetan Plateau Using a Simple Radiation Model, Adv. Atmos. Sci., 29( 4), 675-689, 2012.

Albedo at NC is 0.183 derived from the reference (Zhao et al., 2011).

Zhao X., Peng B., Qin N., Wang W. (2011), Characteristics of Energy Transfer and

Micrometeorology in Surface Layer in Different Areas of Tibetan Plateau in Summer ( in Chinese), Plateau and mountain Meteorology Research,31(1), 6-11, 2011.

6. Line 200: why scaled DSR to 1400 W m$^{-2}$, DSR is net downward shortwave radiation, rather than total solar radiation.

Reply: DSR means downward shortwave radiation, not net downward shortwave radiation. We just adopted 1400 W m$^{-2}$ according to Duchon and O'malley (1998) and Long and Ackerman (2000). It only favors for a clear presentation of the normalized and observation DSR together in the same figure.

7. In addition, the paper would be greatly enhanced with additional proof reading to improve the quality of the written English.

Reply: The manuscript has been extensively revised according to reviewers' comments and suggestions. We tried our best through additional proof readings to eliminate grammar errors.

---

## Author Response (AR2)

Reply to the first reviewer.

General Comment:

The change of land-atmosphere interaction on the Tibetan Plateau has a significant impact on the climate of China and even the whole world. However, due to the high altitude and harsh climate condition, there are few observations and a poor applicability for the empirical models parameterized in other regions. This paper used the measurements with the high temporal resolution to calibrate the parameterization of empirical models for downward longwave radiation, to improve their applicability in the Tibetan Plateau. This paper is expected to provide a useful reference for scientists working on the land surface energy balance over the Tibetan Plateau. To make this paper suitable for publication, please address satisfactorily the specific comments I have appended below.

Reply:

We greatly appreciate the reviewer's opinions on our submission and the manuscript was revised according to these valuable comments and suggestions.

Major Comments:

(1) The biggest flaw in this article is the lack of validation. The author used observation data to optimize the parameterization of the data, but did not use other observation data to conduct effective verification. The dataset used to optimize models were not independent from the dataset to verified models, which seriously affected the credibility of the validation.

Reply:

We partly agree with this comment. The parameterization was established and validated by using 10-fold cross-validation method, which is widely used in the literatures. As suggested in the literature, this method shows the skill of a regression model on unseen data and thereby would be expected to result in a less biased or less optimistic estimate of the model skill than other methods, such as a simple train/test split (James et al., 2013). This means that the parameterizations are, to some extent, validated by this process, although it is not independently assessed.

In order to clearly show the performance of our localized parameterization, we used independent DLR measurements at Lhasa (91.1°E, 29.9°N, 3649 m ASL) during summer in 2012 to verify our model in the revised manuscript according to your suggestion. All clear-sky, cloudy-sky parameterizations including our models were validated and the bias and RMSE are shown in following figure.

[Figure]

[Figure]

Fig.1 BIAS and RMSE for the LDR parameterizations using (a) the published clear-sky model and Eq.(3), and (b) cloudy-sky parameterizations and Eq.(5).

Comparing to the existed parameterizations, the Eq.(3) and Eq.(5) have the smallest BIAS (both less than 2 W·m$^{-2}$) and RMSE (Eq.(3)'s is less than 5 W·m$^{-2}$ and Eq.(5)'s is less than 25 W·m$^{-2}$) at Lhasa. This clearly shows that our localized parameterizations produce the best results.

(2) The change of cloud cover during the day is analyzed based on the downward shortwave radiation, but the change of cloud cover at night is not involved. The authors should examine whether there are significant differences in the amount of cloud cover over the Tibetan plateau between day and night.
Reply:
According to Sato et al. (2007), there are diurnal cycle of clouds over Tibetan Plateau. However, since we mainly calculate cloud fraction from daytime shortwave radiation measurements, DLR during nighttime is not considered in this research and thereby needs further study in future. We did discussed this issue in the section of discussion.

Sato, T., Miura, H., and Satoh, M.: Spring diurnal cycle of clouds over Tibetan Plateau: Global cloud-resolving simulations and satellite observations, Geophysical Research Letters, 34, 10.1029/2007gl030782, 2007.

(3) The main content of this paper is to adjust the parameters of the empirical model so as to improve its performance in the Tibetan Plateau. However, the process of parameter optimization is not introduced in detail in this paper. We want to guide whether there is some physical significance to the parameters adjustment, or what is the selection criteria for you to finally choose those parameters, and whether those parameters are optimal or not.
Reply:
Selection of variables in the DLR parameterization is generally dependent on physical consideration, for example, air temperature and water vapor dominantly determine clear-sky DLR, however, cloudy DLR is substantially influenced by clouds (cloud cover, its bottom height etc.), so nearly all DLR parameterizations use these variables as dependent variables. This is the basic rationale in the DLR parametrization.

It should be noted that the coefficients associated with these variables in the parameterizations vary substantially in different geographical locations that are characterized by different climate. In other words, the values of the parameters, to some extent, are representative of the average atmosphere condition, and therefore locally corrected parameterizations would produce better results and higher time resolution would offer more accurate estimations (Grubler et al., 2012). This is exactly why the DLR parametrizations should be corrected in different places by using local measurements.

The coefficients in the references (Table 2) were originally calibrated and determined in different geographical locations and therefore not be the optimal values for the TP, which is clearly demonstrated by their large systematic and random errors. Thus we take use of 1-minute clear-sky DLR samples to locally calibrate the parameters of these parametrizations. We use 10-fold cross-validation method to determine the parameters. This is a widely used method to estimate the skill of a regression model on unseen data. It is expected to result in a less biased or less optimistic estimate of the model skill than other methods, such as a simple train/test split (James et al., 2013). All the data was randomly dividing into 10 groups of approximately equal size, the coefficients are computed by using 9 groups as the training set, and the remaining 1 group is used as validation. This procedure is repeated 10 times to get the representational value of coefficients (with the lowest test error). Whether those locally corrected parameters are optimal is determined by the validation that demonstrated that our locally corrected parameterizations were the best.

Gubler, S., Gruber, S., and Purves, R. S.: Uncertainties of parameterized surface downward clear-sky shortwave and cloudy-sky longwave radiation, Atmos. Chem. Phys., 12, 5077-5098, 10.5194/acp-12-5077-2012, 2012.

Minor Comments:
(1) Table 4 and table 5 should be combined into one table for easy comparison.
Reply:
Done, thanks.

Table 4. Ordinary and locally fitted cloudy-sky DLR parameterizations

| Reference | $DLR_{cld}$ Parameterization | Ordinary Parameters | Locally Fitted Parameters |
|---|---|---|---|
| Maykut(1973) | $(a + b \times CF^c)\sigma T^4$ | a=0.7855 b=0.000312 c=2.75 | a=0.85 b=0.01 c=3 |
| Jacobs(1978) | $(1 + a \times CF)DLR_{clr}$ | a=0.26 | a=0.23 |
| Sugita(1993) | $(1 + a \times CF^b)\ DLR_{clr}$ | a=0.0496 b=2.45 | a=0.2 b=1.3 |
| Konzelmann(1994) | $(1 - CF^a)DLR_{clr} + b \times CF^a\sigma T^4$ | a=4 b=0.95 | a=3.5 b=1 |

(2) Equation 5 is a very interesting highlight of this paper. In previous studies, it was believed that the cloud base height had a very significant impact on the change of downward long-wave radiation, but most empirical models did not consider the cloud base height due to the lack of observation. However, there are limited introduction and verification for the Equation 5. Therefore, I hope the author can add some graphs and tables to introduce this part in detail.

Reply:

Thanks, we revised this paragraph.

A close relationship between CRE and CBH under overcast conditions over the TP is presented in Fig 6. Compared to Viúdez-Mora (2015) results derived at Girona, Spain, a mid-latitude site with low altitude, CRE over the TP is generally lower by 5~10 W·m$^{-2}$. This is likely because clouds over the TP with the same CBH as that at Girona have relatively lower temperature, thereby producing lower radiative effect on DLR. CRE generally decreases as CBH increases. The result agrees with the expectation since CBH influence on DLR should decrease as CBH increases as a result of increasing water vapor effects on DLR. CRE is about 70 W·m$^{-2}$ for clouds < 1 km and decreases to ~40 W·m$^{-2}$ for clouds at 3~4 km in TP. The decreasing rate of CRE with CBH is estimated to be -9.8 W·m$^{-2}$·km$^{-1}$ over the TP that agrees with model simulations (Viúdez-Mora et al., 2015).

Since CBH effect on overcast DLR is apparent, we introduced a modified parameterization to consider CBH effect on DLR under overcast conditions as follows. The calculated DLR$_{\text{ovc}}$ (DLR$_{\text{ovc}}^{\text{cal}}$) is equal to DLR$_{\text{clr}}$ times 1.23 due to CF of 1 in Eq. 5 that is irrelevant to CBH. On the other hand, the measured DLR$_{\text{ovc}}$ (DLR$_{\text{ovc}}^{\text{obs}}$) is closely related to CHB as shown in Fig. 6, so we calculated the ratio of DLR$_{\text{ovc}}^{\text{obs}}$ to the corresponding DLR$_{\text{ovc}}^{\text{cal}}$ that showed a linear relationship to CBH, as shown in Fig.7. A linearly fitted equation was then established between this ratio and CHB. DLR$_{\text{ovc}}$ is finally parameterized as follows.

$$\text{DLR}_{ovc} = 1.23 \times \text{DLR}_{clr} \times (1.07 - 0.046 \times \text{CBH}) \qquad (6)$$

Where CBH has unit of km. The bias and RMSE of Eq. (6) between measurements and calculations are -2.15 W·m$^{-2}$ and 19.79 W·m$^{-2}$, respectively, which are substantially lower than that of Eq. (5) (10.3 W·m$^{-2}$ and 21.4 W·m$^{-2}$) under overcast conditions. This indicates a remarkable improvement in the estimation of DLR under overcast conditions by introducing CBH to the DLR parameterization, therefore, introduce of such instruments as ceilometer to measure CBH is highly significance for studying cloud's impacts on DLR.

(3) Fig 3 shows that, compared with Zhu's research results, the two are mainly the difference in intercept, which mainly depends on the calibration of initial value and has no great significance for the study of variabilities of DLR.

Reply:

We replied to this concerns in our first round review process, which may be overlooked by the new reviewer.

Difference in the calibration is likely one of potential causes for this systematic difference in clear-sky DLR parameterizations, but we tend to consider potential cloud contamination on the clear-sky DLR samples plays a dominant role in this systematic difference. Cloud contamination of clear-sky DLR samples would be expected if we use naked-eye observations of cloud cover every 1-3 hours (just as Zhu's paper). Human observation represents an instantaneous snapshot of the sky dome that is not likely to represent sky conditions within 1-3 hours since clouds show dramatic temporal variability. More importantly, this systematic overestimation (25 $W \cdot m^{-2}$) is much larger than the expected uncertainty of DLR measurements (2.5% or 4 $W \cdot m^{-2}$) (Stoffel, 2005). Furthermore, comparison of cloudy DLR parameterizations between this study and Zhu et al. (2017) showed a good agreement.

[Figure]

Fig.2 Comparison of our locally corrected cloudy DLR parameterization and that in Zhu et al. (2017) to measured DLR.

Reply to the second reviewer.

1. The manuscript lacks innovation point.
Reply:
    We respect the reviewer's time and efforts on evaluating our manuscript but we cannot agree with this simple claim without any arguments. We clearly stated our innovation in the last paragraph of section introduction.
    This study makes progress in the following aspects as compared to previous studies: 1) clear-sky discrimination and cloud fraction estimation are based on 1-minute DSR and DLR measurements that are objective in nature; 2) misclassification of cloudiness into cloud-free skies would be minimized by adopting strict cloud-screening procedures based on 1-minute DSR, DLR and Lidar measurements; 3) potential effects of CBH on DLR are also investigated. Localized parameterizations of clear-sky and all-sky DLRs are finally achieved, which would be expected to improve DLR estimations over the TP.

2. The calibrated empirical algorithms can not be demonstrated that they also work well at other sites.
Reply: Based on reviewers' comments, we used independent data at Lhasa to validate the parameterizations in the revised manuscript, which clearly shows the best performance of our parameterizations.

3. It is different between the cloud fraction by trained human observers and the radiation-based cloud fraction. They are two different variables.
Reply: We agree with the reviewer's opinion on the difference in cloud fraction derived from human observation and radiation measurements, but frankly speaking, we do not get the key point of this opinion. We argue that human observations seem not suitable for the DLR parameterization under cloudy conditions. The first reason is that the observations are subjective in nature. Second, human observations are made every 1~3 hours that are not enough to capture dramatic variation of sky conditions. Therefore, we use 1-min DSR measurements to deduce sky condition that is then used in the DLR parameterization.

[revised manuscript text omitted]

measured (black line) and calculated (dotted black line) downward shortwave radiation and its 21- min standard deviation (grey line), (b) measured downward longwave radiation and 21-min standard deviation and (c) MPL backscattering coefficient and the cloud base height.

[Figure]

Fig. 2. RMSE and $R^2$ for the clear-sky DLR parameterizations using original (a) and locally calibrated (b) coefficients.

[Figure]

Fig. 3. Scatter plots ofmeasured clear-sky DLR data from as a function of calculations
by the Eq.(3) this study (blue dots) and the Eq.(4) by Zhu et al. (2017) (red dots). The
dash black line is the 1:1 line.

[Figure]

Fig. 4. RMSE and $R^2$ for the cloudy-sky DLR (DLR$_{cld}$) parameterizations using the
original (blue) and locally calibrated (red) coefficient.

[Figure]

Fig. 5. BIAS and RMSE for the LDR parameterizations using (a) the published clear-sky and Eq.(3), and (b) cloudy-sky parameterizations and Eq.(5).

[Figure]

Fig. 6. Distributions of cloud radiative effect against measured cloud base height are represented by box plot (the blue box indicates the 25th and 75th percentiles, the whiskers indicate 5th and 95th percentiles, the red middle line is the median). The black circles line and the black triangles is mean values of cloud radiative effect over TP in this study and in midlatitude site (Girona, Spain) in Viúdez-Mora(2015) respectively.

[Figure]

Fig. 7. Distributions of the ratio of observed DLR and calculated DLR by Eq.(5) under
overcast condition against measured cloud base height are represented by box plot (the
blue box indicates the 25th and 75th percentiles, the whiskers indicate 5th and 95th
percentiles, the red middle line is the median, the black plus sign is the mean). The
black triangle line is the fitting line.

---

## Author Response (AR4)

Reply to the first reviewer.

This manuscript evaluated eleven different parameterization methods in Tibetan Plateau using three ground stations. I don't see any errors as all eleven methods are well-established and have been extensively evaluated otherwise else.
Reply: Thanks.

My questions are as follows.
Lines 115 -117: Zhu et al. (2017) evaluated 13 clear-sky and 10 all-sky DLR models based on hourly DLR measurements at 5 automatic meteorological stations. What are the differences between your work and Zhu's work in 2017?
Reply: Our major point is that clear-sky DLR parameterization may be seriously impacted by clear-sky data samples that are very likely contaminated by cloud residuals if human observations of cloud or hourly DLR measurements are used as the unique criteria in selecting data samples. Our result (Fig 3) clearly showed that clear-sky DLR in the previous studies was very likely overestimated by cloud residuals, which would significantly affect studies that take the clear-sky DLR estimation as their prior requirement, for example, cloud DLR forcing. Moreover, we studied the relationship between cloud base height and DLR that has never been investigated in the TP before. We consider these are our original contributions to our understanding of DLR parameterization in the TP. This research would be not possible if a comprehensive measurement project had not been performed. As one of important parts of a cooperated field campaign, the state-of-the-art pyranometer and pyrgeometer with ventilation and heating system are used to respectively measure downward shortwave and longwave radiation with 1-minute resolution, in addition, Lidar measurements provide much more information about clouds than before. To our best knowledge, installation of radiometers and Lidar side by side has never been performed, furthermore, 1-minute measurements are very rarely reported in the TP. These should be our novel aspects of experimental method, which indeed favors for our DLR parameterization study.

Lines 188 191: Can you use the actual SSA value, for example from any satellite product, instead of one mean value for three station? Surface albedo varies with time. Can you use any satellite product or measure it in situ?
Reply: There are few satellites providing aerosol SSA product except OMI. Given aerosol loading is very low, it is nearly impossible to retrieve precise SSA from satellite, therefore, we used a mean climatic value of SSA in study.
       We did not measure surface albedo in TP and then used surface albedo observation results in same sites and periods from other researchers. Because our observations cover a short period, temporal variation of surface albedo is not likely a key issue in this study.

Reply to the second reviewer.

The manuscript evaluated and locally calibrated 11 clear-sky and 4 cloudy DLR parameterizations using high temporal resolution radiation measurements over TP in summer months. Three methods were combined to discriminate clear sky from clouds, which play an important role in improving the accuracy of parameterizations. The influence of CBH on DLR under overcast conditions was analyzed and a parameterization considering CBH was introduced. The topic is of sufficient interest to the communities of study on solar modelling and climate change. I recommend this paper for publication after revision.

Reply: We greatly appreciate the reviewer's opinions and revised the manuscript according to your valuable comments and suggestion.

Comments:

1. "W•m-2", "Wm-2", and "W/m2" appear in the manuscript and they should be unified.

Reply: We thoroughly revised the manuscript and unified them into "$W·m^{-2}$".

2. Line32: Is the overestimation of clear-sky DLR found in one pervious study or some studies? The statement should be changed to "in one previous study" or "in previous studies".

Reply: Done, thanks.

3. Line 56: If "those remote regions" refer to some particular regions?

Reply: No, we just mean regions without surface observation. We corrected "those" into "some" in manuscript.

4. The abbreviation "T" refers to screen-level temperature in Line 64 and air temperature in Line 295. Different abbreviations should be used.

Reply: Thanks, we corrected this content in manuscript.

5. Line 97: Two predicate verbs appear in one sentence.

Reply: Thanks, we corrected this problem.

6. Line 105: Change "of highly significance" to "of high significance" or "highly significant".

Reply: Thanks, we change it to "of high significance".

7. Line 120: Change "root mean square" to "root mean square error".

Reply: Done, thanks.

8. Line 151: Change "make it having" to "make it have".

Reply: Done, thanks.

9. Line 157: If you want to express "side-by-side"?

Reply: Yes, thanks, we corrected the word into "side-by-side" in manuscript.

10. Line 161: What does "the dataset" refer to?
Reply: "the dataset" means all data we used in this article, we added explanation in the manuscript.

11. Line 187-188: The machine model of Cimel sunphotometer is usually expressed as "CE-318".
Reply: Thanks, we changed "CIE-318" to "CE-318" in manuscript.

12. Line 188-189: Why did the authors adopt the same Angstrom wavelength exponent and Angstrom turbidity in NQ as that in AL? Please confirm the rationality. In addition, the rationality of adopting mean SSA in Lhasa should also be explain.
Reply: NQ and AL have similar altitude (4507 m ASL and 4287 m ASL) and climate (relatively dry) in three sites. Therefore, using Angstrom wavelength exponent and Angstrom turbidity data in AL rather than that in NC is rational.

Considering the extreme low aerosol loading in TP, we simplified the influence of aerosol on $DSR_{cal}$ by using climatic values. Lhasa, which has long-period SSA observation, can provide relatively stable climatic value of SSA in TP. So we decided to use the mean SSA at Lhasa in this study.

Thanks for the suggestion, and we added this content in manuscript.

13. Line 189: Units of latitude and longitude should be added.
Reply: Thanks, we found that we mentioned the latitude, longitude and altitude (with units) of Lhasa in Line 164, so we deleted this content in Line 189.

14. Line 228: Change "an abruptly changes" to "an abrupt changes".
Reply: Done, thanks.

15. Line 262: Change "Previous studies suggests" to "Previous studies suggest".
Reply: Done, thanks.

16. Line 283-285: It is suggested to explain the bad performance of parameterization proposed by Swinbank (1963) and Idso and Jackson (1969).
Reply: We added this content, thanks for suggestion.

17. Line 371: References are needed.
Reply: Done, thanks.

18. Line 388-389: Change "introduce" to "introduction" and "significance" to "significant".
Reply: Done, thanks.

[revised manuscript text omitted]